# The Indians and Major Studies in New Spain: Monarchical Politics, Debates, and Results

Rodolfo Aguirre

Instituto de Investigaciones sobre la Universidad y la Educación, Universidad Nacional Autónoma de México, Ciudad de México 04510, Mexico; aguirre_rodolfo@hotmail.com

**Abstract:** This article studies some stages and debates about the access of New Spain's Indians to major studies: The discussion about their mental capacity in the 16th century, the impulse of Carlos II to the indigenous nobility in the 17th century, or the reticence in the Royal University of Mexico and the Church to their acceptance in the 18th century. It also analyzes the responses given by the Crown to the interest of the Indians elites in superior studies, degrees and public positions, protected by their rights as free vassals of the kingdom and as nobles, comparable to the Spanish nobility. Despite the insistent resistance of sectors of the colonial government and society to the rise of Indians, they firmly defended, in the 18th century, the rights and privileges granted to them by the monarchy since the beginning of New Spain, thereby achieving their entry into the university, colleges, and clergy.

**Keywords:** Indians; higher studies; New Spain; monarchical politics; university degrees; clergy



## 1. Introduction

In the historiography of the 20th century the issue of access of the Indians of colonial Spanish America to major studies was addressed as somewhat anecdotal or exceptional (Cuevas 1922, pp. 453–57; Cuevas 1924, p. 110; Ricard 2013, p. 289; Gonzalbo 1990, p. 170). However, in the last three decades some works have proposed that this access was part of a complex and longstanding process of connecting the nobility to the Hispanic regime (Estenssoro 2003; Menegus and Aguirre 2006; Aguirre 2006; Alaperrine 2007; Lundberg 2008; Martínez 2012; Cobo 2012; Carrillo 2012; Menegus 2013; Decoster 2015; Glave 2018). It was, therefore, not only an academic issue, but also a political and social one, which led to various debates, over time, focused on the level of knowledge that the Indians were supposed to have. The issue was discussed since the early 16th century and was also linked to the question of whether or not to form an Indians clergy.

Although opposition would be constant, Indians had supporters among the regular clergy as Fray Bartolomé de las Casas, some bishops and viceroys, who considered it right to give them an education similar to that provided to the Spanish. For other authorities and influential characters, like the Dominican Fray Domingo de Betanzos or the encomenderos, however, it was enough to educate them only in the Christian doctrine as well as in the reading and writing of Castilian language. For its part, the Crown never established a broad program of creation and provision of schools to educate all Indians. Instead, it commanded the authorities of America to do their best to settle them by looking for resources, but without any engagement from the royal hacienda. Consequently, in New Spain there were several schools run by friars, clergymen, and Jesuits, which had different ranges, depending on local conditions and financial resources available. Most were for teaching reading and writing; others also educated Indians as tailors, blacksmiths or carpenters, to meet the needs of the Spanish colonists.

Regarding the indigenous nobility, their education received special attention. Since the 1520′s it was proposed to educate them with more specific European knowledge, not only by the evangelizing friars, but also by the Crown, since it was considered that the Spanish regime needed it for obtaining political control and taxes from the population. Although

there were debates among theologians and jurists of the Hispanic world on the ability and rationale of Indians, the nobility was treated differently because of its importance to the new political order: they were excluded from paying taxes to the king, they could use weapons and horses, and the legal figure of the *cacicazgo* was established for them, which included rights and privileges over land, labor and tributes of the Indians, the title of "*cacique*" for life, which was hereditary, as well as having a coat of arms, as the Hispanic nobility (Menegus 2005). Charles V ceded to the friars the vanguard of evangelization, control of the instruction and education of the noble Indians, now called "*caciques*". The friars were also interested in educating them as part of their alliance with the indigenous nobility to guarantee human and material resources for their convents. Their children began to learn Spanish in the friars' schools and colleges.

Other important officials and institutions (bishops, clergy, viceroys, the Royal University of Mexico and Jesuits, for example) have intervened in favor or against the inclusion of Indians in the various degrees of knowledge: first letters, Latin, philosophy, canons, laws or theology. Their arguments included social, cultural, political, and even theological issues, indicating that the question involved complexity and no decision was easy. The Crown was generally in favor of granting them major studies, although in the 16th and a good part of the 17th centuries it did not dictate any concrete measures to achieve this. It was only in the 1690s that Charles II firmly promoted this policy in the Indies.

This article seeks to study some of the stages and discussions surrounding the access of Indians to higher studies in New Spain, on the one hand, as well as the response of the Crown to the interest of the Indians elites in this knowledge, the ranks and public positions, protected by the figure of "free vassals" of the kingdom and "nobles", comparable to the Spanish nobility, on the other. To this end, a methodology based on the historiographic concept of the "long term" has been used, (Burke 1999, pp. 40–1) because it is considered that the problem studied here was part of a process that initiated in the 16th century and had its culmination in the 18th century. Likewise, tools from the political and social history of law have been used (Lira 1994, p. 42); specifically, from a reading of the royal cédulas and other legal acts on the Indians people of New Spain in order to analyze the political and social events that explain their birth, as well as the agents that promoted them, or opposed their fulfillment. The study of the legislation on the education of Indians has been enriched with the speeches, discussions and debates of significant figures who have intervened in the matter, such as Friar Bartolomé las Casas, the archbishops of Mexico, Friar Juan de Zumárraga, Friar José Lanciego Eguilaz, and Manuel Rubio Salinas; the viceroys Antonio de Mendoza, Luis de Velasco "El Viejo" or Luis de Velasco, "El Mozo". Some of their writings and correspondence from printed treaties have been recovered, as well as reports and mail from important archives of the Spanish Empire.

In accordance with the objectives of the special issue, the research seeks to demonstrate that, despite fierce resistance from various sectors of the colonial government and society to the rise of Indians, they have taken into their hands the promotion of the rights and privileges that the monarchy granted them since the 16th century in order to gain entry to the university, colleges, and clergy in the 18th century. The Spanish Crown considered the Indians, especially the nobles, as free vassals who held prerogatives according to their social status, with the right to be heard and to receive justice. As part of this treatment, a precedent of the concept of citizenship discussed in the Cadiz courts of 1812 (Salvatto and Carzolio 2015; Ávila 2011, p. 1455), the kings considered that Indians were entitled to receive the same opportunities for higher education as Spaniards. The literate Indians of the 18th century, almost all of whom came from the nobility, were able to access specialized knowledge previously restricted only to Spaniards, and with this they achieved higher-ranking occupations and positions in society. For the Crown, the access of the Indians to these positions allowed it to integrate them into the Hispanic regime of government in the New World as well as to reinforce their loyalty to the monarchy. To this end, this article has been divided into four sections corresponding to decisive stages. The first goes from the conquest of Mexico to the third Mexican provincial council of 1585, a period

focused on the debate on the type of education and knowledge that the conquered Indians should receive. The second covers the last third of the 17th century, when, thanks to the vindictive arguments of Peru's *caciques* in Madrid, they managed to get Charles II to issue concrete measures to facilitate the Indians access to colleges, scholarships and university degrees. The third section studies the arrival of the Indians to the colleges and universities of Mexico in the 18th century and the reactions that this event caused in the authorities. Finally, the fourth section analyzes the admission of Indians to the clergy and some parish posts, the main destination sought by them, as well as the opinions of church authorities on the subject.

## 2. From the Conquest to the Third Mexican Council

The Mexican capital of Tenochtitlán fell in 1521 under Cortés' army and three decades later, in 1551, Charles V established the first university in New Spain, with the order to allow the entrance of Indians. Nevertheless, in 1585, the third Mexican council substantially limited their access to the clergy. These decisions, of 1551 and 1585, illustrate the diversity of stances that marked the discussions on the admission of Indians to higher studies. One of them was the debate on the nature of Indians and their rationale.

Many European colonizers considered that Indians were little or not at all rational men, who could be enslaved, and others, more condescending, as neophytes, who were not yet Christians and, therefore, there would be no justification for educating them or allowing them to join the clergy (Bosh 1987, p. 27; Beaudeau 1990, pp. 37–73). However, in 1523, Charles V declared Indians free, after having consulted " . . . theologians, religious and people of many letters . . . " (De Encinas 2018, book 4, f. 248), for which they were to receive the same treatment and education as the Castilian vassals.

However, the issue continued to spark debate among prominent theologians and jurists. Bartolomé de las Casas and other famous 16th-century thinkers argued that the Indians were neither barbarians nor irrational, even though they were not Christians and, therefore, no one had the right to enslave or mistreat them. Neither did they accept the need to convince them in order to civilize them and argued that they should not lose their property or their freedom (García 2000, pp. 63–64). On the contrary, they proposed that Indians should, voluntarily, know the true religion and even if they were to reject it, this did not mean that they should be conquered with arms. Las Casas, whose ideas and writings were broadly known, even by the indigenous nobility of the 18th century, advocated that the Indians had natural and moral qualities superior to those of the Spaniards (Maestre 2004, p. 122). Moreover, he lauded the level of civilization that the Indians enjoyed before and after the conquest:

> [ . . . ] they are all naturally very fine, alive and clear and very capable of comprehension [ . . . ] endowed with the three kinds of monastic prudence, according to which man knows how to govern himself; economic, he knows how to govern his house and political, he orders and arranges to govern the city [ . . . ] they are laborers, artisans, people of war, rich men, the religion, temples, priests and sacrifices, judges and ministers of justice, and governors, habits and in all that concerns the acts of the understanding and the will [ . . . ] (De Las Casas 1909, pp. 683–84)

At the same time of these discussions in the first half of the 16th century, another process started: the adaptation and negotiation of the indigenous nobility with the new Hispanic regime. Not only did the descendants of Mexico's most powerful lineages, Texcoco and Tacuba, did so, but also most of the natural rulers, who aligned themselves with the monarchic power and its agents, headed by Hernán Cortés, acknowledging Charles V as the new supreme ruler and providing him with a good part of the tax. Furthermore, they agreed that their children go to the new schools of the friars to be educated and tutored in the new religion, such as the famous school founded by Friar Pedro de Gante (López 2016) or the imperial college of Santa Cruz de Tlatelolco (Gonzalbo 1990, pp. 111–33). This shows the early interest of the indigenous nobility in joining Christians

and becoming loyal vassals of the Spanish Crown, facilitating the political transition. Regarding these issues, Las Casas explained in 1552 the Novohispanic experience regarding the major studies of the indigenous nobility in this initial stage:

> [ . . . ] we have no less experience in being these very ingenious people in whom they have been taught until today [ . . . ]. As for the other liberal arts, such as grammar and logic, which up to now have been taught to them, no one is unaware of those who have been in New Spain, secular and less ecclesiastical and religious, of how much they are used and how good Latinos in particular are, and that is what they have been trained in most. From where it is clear that each and every time they want to give them study and doctrine in the other sciences, they will come out good and perhaps very marked. (De Las Casas 1909, p. 166)

However, the indigenous nobility was not unconditional towards the monarchy since they asked in exchange for power, government positions, taxes and social recognition. As a response, the Crown gave them recognition, privileges, rewards and even allowed them to travel to Spain to manage their demands before the Council of the Indies. Several Indian nobles even requested a personal audience with the monarch. Between 1527 and 1530, Tlaxcalan and Mexica delegations were received by the emperor (Díaz 2010). In New Spain, noble Indians acceded to the new positions of governors and perpetual rulers in Indians villages.

Likewise, since the 1530s, the relevance of training Indians priests was discussed. The Franciscans argued that since they were free vassals and former lords of the land, they should have the same privileges as the Spanish. The opposite position, defended by encomenderos and colonizers, considered that Indians, being a conquered people, should not have such benefits (Menegus 1987, pp. 83–89). A very powerful Dominican in the first period of evangelization, Friar Domingo de Betanzos, stated that Indians lacked the capacity to understand the faith. However, this was contested by the bishop of Santo Domingo, Sebastián Ramírez de Fuenleal (Gonzalbo 1990, p. 107).

Despite these opinions, some clergymen and laymen encouraged schools and colleges to form an Indians clergy. The Franciscans, aided by the first bishop of Mexico, Fray Juan de Zumárraga and the first viceroy, Antonio de Mendoza, founded the imperial college of Santa Cruz de Tlatelolco where the children of *caciques* were taught Latin and humanities, hoping to find in them a vocation for the Christian clergy, but this was not the case at that time. As a result, those who opposed the possibility of granting them higher education had sufficient arguments to end the trial and, consequently, that college was only intended to teach the doctrine and the Castilian language. Tlatelolco was the greatest effort in the 16th century to give upper schooling to an Indians elite (Gonzalbo 1990, pp. 111–33). In other colleges of the regular clergy, contemporary with Tlatelolco, Indians were excluded from the priesthood (Gonzalbo 1995, pp. 287, 290). In the case of the Dominicans, although at the beginning they indeed accepted some Indian novices, they soon banned them (Méndez 1990, p. 18).

The counterpoint to this tendency originated from several voices, like that of the bishop of Tlaxcala, Fray Julián Garcés, who penned a letter to Pope Paul III defending the natives and refuting those who thought them incapable and irrational: " . . . they write in Latin and in Romance better than our Spaniards, and those who study Latin and Spanish among themselves are no less effective than we are." (Gonzalbo 1985, p. 76) Shortly afterwards, Paul III issued the bull *Sublimis deus* in 1537 in which he declared that the Indians had a soul and were rational (Sempat 1998).

Another key moment in favor of Indians came with the New Laws of 1542, which forbade slavery and the encomiendas, reaffirming their status as free vassals of the Crown (De Encinas 2018, book 4, f. 263). These laws were the outcome of a gathering of notable theologians and jurists summoned by the emperor in Valladolid, where Fray Bartolomé de las Casas was a major figure (García 2000, pp. 80–81).

Moreover, at the beginning of the 1540s, the War of the Mixtón, a very important uprising of thousands of Indians in western New Spain (García 2010, p. 238), challenged

the nascent Hispanic regime and fuelled the fear of a general rebellion that would end up with the Spaniards. Probably, in the same proportion, the refusal to allow Indians to have any kind of education, much less the so-called elders, rose.

In this context of debates, declarations of the Crown, a New Hispanic environment that still allowed the enslavement of Indians and where the *encomenderos* fought to impose their interests, it was important that a university project arose where they were included. As far as we know, the first request to the Crown to establish higher studies in Mexico was by the accountant of the royal estate, Rodrigo de Albornoz, in 1525, who suggested a college for the children of *caciques* where they could learn Latin, philosophy, and other disciplines " . . . so they could become priests, who would take more advantage of what came out of them and make more fruit than fifty of the Christians . . . (Méndez 1990, p. 58) It is remarkable that a civilian would care to create an Indians clergy only four years after the conquest of Mexico. In the 1530s other figures resumed this idea, such as the bishop of Mexico, Friar Juan de Zumárraga, who viewed the university as a seedbed for the creation of a local clergy. In 1536 he ordered his procurators in Spain to solicit the foundation of a university, in view of the great need for the conversion of Indians and to rectify errors of doctrine, adding that if in Granada, where there were much fewer neophytes than in New Spain, the king had already founded one, with more reason in Mexico " . . . in which all the faculties that are usually read in other universities and teach, and above all arts and theology, are read, because of this there is more need." (Méndez 1990, p. 65) In order to support it he proposed to provide it with the tax of some tribe of Indians.

The viceroy Antonio de Mendoza, in 1537, thought that the university could only be assured with the help of the royal hacienda, as in Granada. Similarly, he considered that the Indians who studied at the college of Tlatelolco were very able to study any subject, for which he suggested to the king to keep on supporting him (Mendoza 1537, quoted in Méndez 1990, p. 108). Two years later, this viceroy also supported a request from the civil chapter of Mexico City to found the university, arguing that this would allow the children of Spaniards to remain in the new world and because there was a good number of students, Spaniard and Indians, who merited access to higher studies. Although the Crown approved the foundation at the time, it didn't provide financial support and only named Juan Negrete as professor of theology (Méndez 1990, p. 74). However, the establishment did not take shape: it did not have its own building, only a professor and no guaranteed income to hire more teachers. It was still far from being an educational institution. For that reason, in 1543, the Mexican *cabildo* insisted on asking for a better economic endowment, demanding a treatment similar to that of the college in Tlatelolco.

In 1546, Prince Felipe requested reports from Viceroy Mendoza on the issue, showing how incomplete the university project was. In 1547, seeking a better endowment, the viceroy and Mexico City received the promise of acquiring the taxes of a commission to sustain the general education (Méndez 1990, p. 88). However, this issue did not go far either, so in the last years of his government in New Spain, Viceroy Mendoza tried new methods: he named more professors and allocated the revenues from some cattle ranches to them. At that time, in 1550, some Franciscans from Yucatan also solicited a university to educate unemployed Spaniards and "mestizos", from which they could form good men for evangelization and no longer depend on Spanish friars; nevertheless, they neglected to include Indians in their request (Méndez 1990, pp. 95–96). Possibly this decision was in line with the negative opinion of Indians derived from the failure of the educational institution in Tlatelolco. Nevertheless, the Crown went on to include them in their university plans, arguing that since the first applications Mexico City had done so: " . . . that Mexico City has often pleaded for a university of all sciences to be founded on that land, where the Indians and the children of the Spaniards would be instructed. (Quoted in Méndez 1990, p. 119).

Finally, the Crown founded in 1551 the Royal University of Mexico and its educational institution of five faculties, through a document that expressed " . . . where the natives and the children of the Spaniards were instructed in the things of the holy Catholic faith

and in the other faculties . . . (Martínez 2006, p. 87), providing it with 1000 gold pesos from the royal hacienda. It is probable that the Council of the Indies, despite the fact that the Mexico City cabildo or the Yucatan Franciscans had solicited it for Spaniards basically, had considered previous opinions that actually included Indians. It is also likely that the insertion of Indians was derived from the policy of the New Laws of 1542 and the Lascasian policy of the Valladolid boards of 1550–1551, to protect and promote Indians. Such inclusion was a successful one for its advocates, although it still needed to be exercised in practice, another difficult task.

For their part, the Spanish considered the new university as an intellectual, social and political project in which Indians had no place (Menegus 1987, pp. 83–89). Viceroy Luis de Velasco, for example, in giving an account of its opening and progress in 1554, praised the benefit made to the young Spaniards but, as far as Indians were concerned, proposed that they not be admitted:

> It seems to religious and learned persons that treat understand them that for the time being it is not convenient to put them in other sciences, and that it is enough for their little being and understanding to know the Christian doctrine, and to persuade them, as it is done, to believe and keep it. They read and write many of them in their own language, and there are great copies of doctrines made by clergymen in the languages of the provinces, approved by the prelates. What matters in raising some of the natives in studies and schools is that, after their studies, they are distributed among the peoples and teach to the natives what they have learned, although some of them have turned out so badly that they would better not have studied. (Cuevas 1914, p. 186)

In Peru there was also a worry at that time about whether or not to encourage Indians to go to university and be trained as Indians priests (De Encinas 2018, book 1, fs. 206–7). However, Indians had no conditions to claim their access to the university. The great epidemics had begun in 1546, decimating many villages. In addition, Philip II set a new tax policy that impoverished many. Likewise, the political power of the *caciques* diminished in favor of Indians cabildos; moreover, they lost land and taxes.

To all this, we must add that in the ecclesiastical institutions, which were already designed for educated Indians, there was also resistance. In the first Mexican Council of 1555, for example, their access to priestly orders was banned, under the argument that they were new to the faith (Pérez et al. 2004, First Council, chapter XLIV). Bishops, religious and other authorities all agreed that Indians should only aspire to be good Christians and tax payers to the king, not literate, much less priests. The same thing happened in the third Mexican provincial council of 1585. There, relevant topics were debated regarding Indians, such as the Chichimeca war, tithe payment, or personal services, with the opinions of theologians and jurists who stood up for a better treatment for them, according to the Crown's policy. Regarding the access of Indians to the priesthood, there continued to be strong reticence, although there was no outright prohibition. A decree established that: "The Indians and the mestizos are not admitted to the sacred orders but with the greatest and most careful election; but in no way those who are noticed of some infamy" (Martínez and García, Berumen Marcela Elisa Itzel and García, Hernández Marcela Rocío 2004, Third Council book I, title IV, paragraph III). Possibly this limited concession was made so as not to oppose the writ of establishment of the university that did include them, as well as the bull *Nuper ad nos* of 1577, which dispensed illegitimacy to Indians, "mestizos" and Spanish priests who knew languages, in order to confront the lack of priests in the evangelization of the Indies (Poole 1981, p. 641). It was not an easy matter to eliminate them, strictly speaking, since there was no canonical impediment for the Indians who were legitimate sons and without any infamy to accede to the priesthood. Thus an argument that became recurrent was to consider them new Christians, to which were added the adjectives of being vulgar, lazy, and conducive to sin and drunkenness, which was already very close to infamy.

The policy of the third Mexican council on Indians' priesthood was replicated years later in Jesuit schools for noble Indians because they did not contemplate major studies but only the study of Castilian and doctrine (Gonzalbo 1995, pp. 153–73), even though some parents considered the possibility of forming clerical Indians. Even Viceroy Luis de Velasco, "El Mozo", asked the king to allow the Jesuits to establish a school for Indians of higher studies in the neighborhoods of Mexico City, to teach Latin, Castilian, and medicine, leaving open a possibility to train priests as well (Cuevas 1914, p. 430). However, there were no results and schools like San Gregorio in Mexico or San Martín in Tepozotlán only educated their students to assist the Spanish clergy in their tasks as acolytes, or to conveniently govern their communities in the future (Olaechea 1958; Zubillaga 1969). At that time, the idea of an Indians clergy was almost forgotten. In the 17th century the Franciscans gave the habit to some of the children of the indigenous nobility, but it was something exceptional (Morales 1973). Hope for the indigenous nobility remained to the decision of the Crown.

### 3. Charles II and the Impetus to the Indigenous Nobility

In the 17th century, the Royal University of Mexico expanded and became consolidated through the interest in the degrees and courses of Spanish students from different cities and towns in New Spain, who characterized it as a corporation of prominent scholars from noble, honorable, and bloodless families, distant from menial and mechanical occupations (Chocano 2000, pp. 157–82). Notwithstanding this idealization, in practice the students already came from a wider social background. At the time of Viceroy Cerralbo, an episode in the Faculty of Medicine in 1634 sparked a discussion about the presence of "mestizo" and "mulatto" students in university classrooms, led by some Spanish students opposed to such a development (AGN, Fondo Universidad, vol. 40, f. 172). This situation was a reflection of the new conformation of the New-Hispanic society, where a greater number of social groups were struggling for social advancement.

The issue was soon addressed by the King's Visitor, Juan de Palafox y Mendoza, who penned new university constitutions, including one intended to regulate the social origin of students and graduates, reflecting the fear of Spanish university groups of a rise in students of other socio-racial qualities. Constitution 246 established the following:

> We order that anyone who has been imprisoned by the holy office, or his parents or grandparents, or has any note of infamy, shall not be admitted to any degree of this university, nor shall blacks or mulattos, nor those commonly called brown Chinese, nor any kind of slave or who has been one: for not only shall they not be admitted to a degree, but neither to matriculation; and it is declared, that Indians, as vassals of His Majesty, can and must be admitted to matriculation and degrees. (González 2017, p. 178)

With this Constitution, the King's Visitor responded to several social concerns of the corporation, but it is important to emphasize the claim of the right of Indians, vassals of the King, to study and receive degrees, as stipulated in the founding charter. While the visitor excluded groups unwanted by the university elite, he did not do so with the Indians. With this, the high minister reaffirmed the monarchic policy of encouraging the social and academic promotion of Indians. Thus, the university continued to be the main space designated by the Crown for Indians to access higher education. However, the problem for this to become a reality continued to be an unfavorable social context since Indians still lived through a constant demographic crisis and the nobility was impoverished and marginalized (Sempat 1989).

At the end of the 17th century, new Crown policies and conditions of Indians caused a major change in expectations. Firstly, the reconstitution of Indians communities and the beginning of their demographic recovery (Carmagnani 1988; Miño 2001, pp. 23–45). Secondly, a more determined policy of Charles II in favor of the indigenous nobility and, thirdly, the willingness of the *caciques* to pursue an ecclesiastical future for their descendants, after almost two centuries of being assimilated into Hispanic culture.

The new stage of the monarchic politics received an important impulse in the second half of the 17th century, when a group of *caciques* from Peru agreed to change their situation and with it, helped their peers from other regions. A process apparently unrelated to the issue studied here resulted in a decree in Madrid that would favor all the indigenous nobility of Latin America.

In Peru, with the legalization of irregular lands of Spaniards, a process known as "Composiciones", from the middle of the 17th century, many Indians and *caciques* were affected by the alienation of their properties. As a result, several of them presented multiple claims and lawsuits at the Royal Court in Lima, but with little success. However, a resolute group of *caciques* decided to travel to Madrid to present their claims to the Council of the Indies. Here, in addition to claiming a solution to their lands, they also claimed their right to obtain the same political and ecclesiastical jobs as the Spaniards and the recognition of their nobility (Glave 2018). This broadening of objectives caused more Andean *caciques* to travel or appoint attorneys in Madrid.

Peruvian pressure paid off in the last decade of the 17th century, not only in their favor but for all the indigenous nobility of the Americas. A decree of 1691 ordered that in the cities, towns, and viceroyalties of New Spain and Peru, schools be established to teach Indians the Castilian language, a requirement for access to government functions (Konetzke 1962, vol. 3, first volume, pp. 11–13). That same year, another cédula authorized the establishment of a seminary college in Mexico City for the secular clergy and that one quarter of its scholarships should go to the children of *caciques*, a measure that would be extended to other seminaries that would be founded in the future (Chávez 1996, vol. I, p. 142). With this, the king demanded other certificates that since the 16th century favored the study of Indians. The Archbishop of Mexico, Francisco de Aguiar y Seijas, was prompt with the order and in 1697 opened a conciliar seminary, stipulating that the scholarships would be given to descendants of *caciques* of the city and the archbishopric (AHSCM, exp. 199/D-II-2, fs. 6v-7). In order to do so, the applicants should prove their indigenous nobility and be pure of blood.

The most important request in favor of Indians is contained in two memorials addressed to Charles II, in 1692, by the prebendary of the cathedral of Arequipa, Juan Núñez de Vela, of "mestizo" origin and Indians descent. A first memorial requested that "mestizos" be allowed to ascend to all positions and honors of the Inquisition, which was accepted (Glave 2018, p. 209). The second was presented in the name of all Indians and "mestizo" people of America and asked that the noble Indians be allowed access to the ecclesiastical dignities, bishoprics, habits of military orders, and "mestizos" to places and functions that required blood cleansing, such as colleges, universities, churches, military posts, chaplaincies, and other jobs of the Royal Service. The Council of the Indies accepted that although there were cédulas that favored Indians, they were not honored, such as those that ordered schools of Castilian, to allocate a quarter of the scholarships to the children of *caciques* in the conciliar seminaries, access to political functions and recommendations of viceroys, governors, bishops, and archbishops in favor of Indians, as was the case with the rest of the vassals of the Indies, to apply for honors and benefits (Konetzke 1962, vol. 3, first volume, pp. 64–66).

The highest moment came with a decree of 26 March 1697, addressed to the viceroys, governors, bishops and archbishops of the Indies. In it, Charles II encouraged the general ascension of the "Indo-Mestizo" nobility to the institutions previously exclusive to Spaniards (AGN, Fondo Reales Cédulas Originales, vol. 27, exp. 11). The document reviewed several earlier provisions that allowed Indians and "mestizos" to enter the clergy and "mestizo" nuns' convents, and then raised the question of whether the Indians could also access, like the Spaniards, the ecclesiastical, governmental, political, and war functions that demanded purity of blood and nobility. The answer of the Crown was that, since before the Conquest there were main and common Indians, and this hierarchy was preserved, the promotion of the natives to Spanish offices should take into account this distinction. The noble Indians were to be equated with the noblemen of Spain, whereas the tributaries with

the common and clean Spanish blood. For the tributary Indians it was enough to open schools of Castilian, while the children of *caciques* were able to do higher studies and obtain a quarter of the scholarships of the conciliar seminaries. The last part of the charter ordered viceroys, audiences, governors, bishops, and cathedral councils to enforce all this.

In New Spain, before the 1697 cédula there were two uprisings that rocked the political and social regime of New Spain: the rebellion of Tehuantepec in 1660 (Castro 2019, pp. 16–17) and the great tumult of Mexico in 1692, which had an important replica in Tlaxcala (Silva 2007). Order could soon be reestablished, providing exemplary punishment to the most rebellious Indians leaders, but the Hispanic regime focused more on the actions, reactions and posture of the Indians ruling elite, a crucial link for political stability. For their part, the indigenous nobility of Mexico City, during the 1692 investigations to punish the leaders of the tumult, dissociated themselves and reiterated their adhesion to the Hispanic regime. Soon after, Archbishop Francisco de Aguiar y Seixas opened the courses of Mexico's conciliar seminary naming three sons of *caciques* as scholarship students. The news spread throughout the viceroyalty and the reaction of the indigenous nobility was immediate, because during the following years more noble Indians began applying for scholarships at other colleges and knocking on the doors of Mexico's Royal University (Aguirre 2006).

## 4. Indians at the University of Mexico and in the 18th Century Schools

The 1697 decree, the result of the Peruvian *caciques*' claim, was known and defended by their peers from New Spain, especially from the center and south of the viceroyalty. Although we do not yet know the way in which this important document was disseminated, it is clear that the indigenous nobility soon became interested in its implementation. It was not the first time, of course, that the Crown decreed cédulas in favor of Indians, since there were many since the 16th century. Why was this one so special for the *caciques*, both Andean and Novohispanic? Glave (2018) gives us important clues: in Peru, many *caciques* claimed and fought, first in Lima and later in Madrid, which managed to bring them together and seek new goals. It is not yet known whether some of the *caciques* from New Spain joined the Andean movement, but it is known that the Arequipa prebendary expressed his willingness to represent everyone in the Americas and even referred to the rights of Moctezuma's heirs. The Andean *caciques* particularly sought out military and government positions.

In New Spain, on the other hand, the indigenous nobility did not fight for military positions but chose the path of higher studies and the priesthood. The *caciques* of New Spain considered the 1697 *cédula* and the scholarships of the seminary colleges an unbeatable opportunity that they should take advantage of. Three were their goals: scholarships to seminary colleges, university degrees, and priestly orders. Although there was no written Church law requiring a college degree for the priesthood, in practice, possessing one was considered an indicator of sufficiency of knowledge (Aguirre 2012). In addition, the Indians showed their knowledge of languages, especially Mexican and Otomi. The indigenous nobility argued that the Crown and the 246th constitution of the university gave them the right to study some faculty. In 1703, the *cacique* Pascual de los Reyes declared his desire to " . . . continue his studies because of my inclination to the sacred ministry of the priesthood, because I am suitable for it, through the privileges and royal laws that we enjoy in which these qualities concur, without anything to the contrary and according to examples". (AGN, University Fund, vol. 44, f. 244)

Discreetly, but with determination, children of chiefs began to arrive at the university gates. The situation of the indigenous nobility at the beginning of the 18th century was different from that of the 16th century: now they were more assimilated into Hispanic circles, had greater support from the Crown and wanted to integrate into the secular clergy. Although there was resistance against it, it did not give up. In 1716, Juan Antonio de los Santos Águila, son of *caciques*, expressed in his request to the rector to graduate from high school: " . . . your lordship will be pleased to order the admission in accordance with the

royal law of the Indies as a statute ... ". (AGN, University fund, vol. 71, exp. 2) Although it was clear that no law prohibited Indians from attending university, the first applications caused confusion. The secretary of the university, responsible for receiving applications, consulted the rectors before enrolling the Indians, something that did not normally happen with the Spanish, unless they had specific news of a social origin prohibited by the statutes. The rectors, with or without reticence, had to adhere to the constitution 246 that allowed the natives access. Closing the way to them could lead to litigation in the Real Audiencia or the Council of the Indies, something they decided to avoid.

Nevertheless, the Indians were scorned and marginalized by fellow students, both in the university and in the schools, because the Spanish students denied having indigenous blood. From the perspective of the doctors of the faculty that governed the Royal University of Mexico, the corporation was only composed of distinguished scholars, far from the "unhappy constitution of the Indians" (AGN, Fondo Universidad, vol. 25, fs. 199v–207v). Likewise, the Indians participated little in the academic life of the university, judging by their relations of merit (Menegus and Aguirre 2006, pp. 263–93). They were not oblivious to this reality and, consequently, they also displayed their noble quality, highlighted in the 1697 charter, but not required by the university, in order to distinguish themselves from the tributary Indians. Moreover, it was clear to them that in the Conciliar Seminary of Mexico, scholarships were to be given only to noble Indians (Aguirre 2012, p. 32).

The presence of Indians in the university was no longer something exceptional and, although they have always been a minority segment, their number gradually rose. Some were even doctors, such as José Antonio Bautista Frías, a priest and lawyer, who in 1770 obtained Bachelor's and Doctor of Law degrees. His case caused doubts from the master school, an ecclesiastical minister authorized to sanction the higher degrees, so he decided to ask the law professors for their opinion on the matter. In his opinion, Professor Nuño Núñez de Villavicencio, designated to study the case, expressed

> The 246th of our constitutions forbids the admission to any degree in this university, whoever has any note of infamy, and blacks, mulattos, and brown Chinese: but expressly states, that the Indians, as free vassals of His Majesty can and should be admitted to matriculation and degrees [ ... ] (AGN, Fondo Universidad, vol. 269, f. 769)

This ruling left no room for further doubt and Bautista Frías was able to obtain a Doctorate in Law, setting an important precedent for other candidates of that social origin.

It is difficult to determine exactly how many Indians students attended the university, since the social origin was not mentioned in the students' records. However, other university papers show that between 1692 and 1724 there were at least eleven (AGN, University fund, vols. 42–46 and 70–71), although it is known that there were Indians who presented themselves as "mestizos" or Spaniards, (Menegus and Aguirre 2006, pp. 216–21) showing altered christening feasts, or reports from convenient witnesses, so their number may be greater. Records of bachelor of arts degrees help to support this hypothesis. The registered number of Indians is 134, between 1711 and 1822, most of them between 1750 onwards (AGN, Fondo Universidad, vols. 167–170 and 293). This figure is also a minimum since other sources from the historical archive of the Conciliar Seminary of Mexico register a greater number of Indians in college, as we will see later.

The records of the university indicate that the schools where Indians studied Latin were not only the conciliar seminaries. Between 1711 and 1732, seven of them enrolled in the conciliar seminary of Mexico, seven in schools in Puebla, three in the Jesuit school of Oaxaca and one in the university. This proportion reflects the general orientation of all the high school graduates, since most studied in the schools of Mexico and Puebla (Aguirre 2002, pp. 25–52).

The number of colleges that admitted Indians increased in the second half of the 18th century. Between 1751 and 1767, the year of the expulsion of the Society of Jesus, there were 11 schools with 36 Indians, distributed in the cities of Puebla, Celaya, San Miguel el Grande, Valladolid, Oaxaca, and Querétaro (Aguirre 2006, p. 87). Later, in the period from

1767 to 1822, 89 Indians from 12 Novohispanic schools were registered (AGN, University Fund, vols. 167–170). Thus, the Indians were educated in the main cities of central New Spain, seeking social advancement and public offices.

It is important to point out the absence of schools in Mexico City in these records. This is probably due to the fact that the Indians who studied in schools in the capital were registered as university students, since they studied in both institutions. The records of the Conciliar Seminar of Mexico show this. At this institution, a little more than two hundred Indians studied between 1697 and 1822, originating from diverse provinces of the archbishopric. The royal students, that is, those who had a scholarship, were 77 (AHSCM, Information on Students 1697–1822). On the other hand, the "portionists "were 117. They were called like that because they used to pay a certain amount of money, or portion, to be able to live and study in the school facilities.

## 5. From Major Studies to the Priesthood

The passage from the classrooms to the priesthood was the favorite option of the noble Novohispanic Indians, who were very close to the Church since the 16th century. Although in this century they were not interested in being friars, they helped widely in the conversion of the people they governed, in the building of churches, chapels and in the support of the ministers. In recognition, they benefited from prerogatives and honors, such as burials inside churches, masses, preferential places in religious celebrations, and important auxiliary positions in the parishes, such as prosecutors, bailiffs, or stewards of religious fraternities (Alberro 2019).

In the 18th century the indigenous nobility was better adapted to the Hispanic regime, was more familiar with its laws and fully shared its values on hierarchy, honor, and privilege. The *caciques* were regarded as the most distinguished vassals of the original population, who had voluntarily ceded their lordship to the kings of Spain and who were a stronghold for the latter to preserve their dominion over the New World. Moreover, Charles II had expressly ordered the diocesan Church to train the Indians clergy in its seminaries and to ensure that they could take up ecclesiastical positions. The *caciques* greatly valued the access to the secular clergy of their sons, not only because of the impulse of the Crown, but also because with the parochial positions, the Indians priests could strengthen the rank of their families as well as their networks of local power.

Yet, the outlook was less promising for Indians in Spanish church circles because there were groups that still considered them to be of lesser social quality. Furthermore, the former lacked the connections and advice needed to embark on a successful religious career. Even if the Indians met the requirements of studies and preparation demanded by the Church, the social values defended by the ecclesiastical hierarchy would tend to exclude them, as was the case in the university and colleges. A notable manifestation in this regard was in 1696, when a group of students and vagrants burned the pillory of Mexico's main square, freeing certain inmates (AGN, Fondo Universidad, vol. 43, fs. 326–328v). In response, Juan Antonio Ortega Montañés, bishop of Michoacán and acting viceroy, ordered the university's rector to apprehend the bad students in order to give public satisfaction to the authorities; but he also ordered to prohibit access to the university to all those students: " . . . who are not Spanish, in consideration of being those who disturb the peace and without any respect for justice . . . " (AGN, Fondo Universidad, vol. 43, fs. 326–328v) The university's faculty of doctors met to discuss the viceroy's order to keep Indians and "mestizos" out of the university, which he accepted without formally expressing any dissent. Then, the rector ordered the publication of an edict that ordered the application of the 246 constitutions, about the quality of the students. However, once the viceroyalty government of the Bishop of Michoacán ended, the Indians resumed the application for courses and degrees at the university without anyone stopping them.

A less strict position was that of Mexico's archbishop, José Lanciego Eguilaz who, in 1715, communicated to King Felipe V the situation of the archbishopric; specifically on the

lower clergy, he expressed his disappointment for their lack of vocation but also for their social origins:

> [ . . . ] I have recognized in my clergy a considerable multitude and great poverty, and what is worse, a mixture of subjects of unknown ancestors, whose pernicious effect results from the indiscretion with which, in bulk, and without distinction, many are ordained in the vacant seat rather than by divine vocation [ . . . ] (AGI, section Mexico, leg. 805, letter to the king of 3 April 1715)

Based on this perception, Archbishop Lanciego sought to achieve important changes through three actions: strengthening the Mexican conciliar seminary, increasing the demand for ordination to the priesthood, and increasing the number of clergymen who are experts in indigenous languages. On this last matter, in the archbishopric there were 48 priests who spoke Nahuatl, 20 Otomi, 12 both languages, six Mazahua, three Huastec, and one Matlatzinca. As part of his strategy, the prelate also promoted the ordination of clerical Indians. In a letter to the king, dated April 1722, Lanciego pointed out that he encouraged them so that: " . . . after practicing our language and the sciences, those who outstand in the conversion and education of those poor Indians would be more helpful than the Spaniards themselves, since love for their own is so natural, and that in this consequence, the learned would also occupy positions in the priesthood (AGI, section Mexico, leg. 805, letter of 13 April 1722). The government of this archbishop stood out, then, for promoting the Indians clergy, despite the criticism he made of the clergy in general.

The Indians reached the priesthood thanks, fundamentally, to their knowledge of native languages. Between 1717 and 1727 there were at least 23 Indians in the archbishopric who were tested to obtain some sacred order. The majority asked to be given ordination as native speakers (AGN, Fondo Bienes Nacionales, leg. 1271, exp. 1). The results were not good for everyone, since nine were rejected for their deficiency in Latin, but not in the language. The 14 who were approved, in addition to their mastery of some language, had barely enough knowledge of Latin and doctrine, which was not so required by the examiners because they considered that with the Indians they should not be rigorists (AGN, Fondo Bienes Nacionales, leg. 1271, exp. 1, f. 170). A few decades later, another archbishop, Manuel Rubio Salinas, made an unfavorable report on the fate of the Indian priests in the archbishopric of Mexico:

> As for the languages, outside of Spanish, many Spanish subjects are ordained as well as Indians and mestizos who are called *cuarterones*. They are assigned by the archbishop, according to the need of their respective peoples, to serve as vicars to the priests [ . . . ] Their instruction is generally limited to grammar and moral matters, and to a perfect understanding of the languages. And, in proportion to their talents, virtue and time they have administered, they are accommodated in curates of their language and in the parishes where their own priests die, until the case of provision arrives and meanwhile they perceive in full the benefits and emoluments of the benefit and pay their assistants. (AGI, section Mexico, leg. 2547, Report of the clergy of the Archbishopric of Mexico of 1764)

Decades later, in independent Mexico, a prominent politician, Lucas Alamán, said in this regard: "The Indians [ . . . ] who were admitted to the priesthood entered schools to learn the ecclesiastical sciences, but in general they were limited to only the knowledge necessary to be ordained and go to run a small priesthood or vicarage, in some distant village and in a bad atmosphere. (Alamán 1990, vol. I, p. 26) These descriptions reveal the marginal place that the Indians held in the ecclesiastical institutions. Even though they were already recognized as a segment of the parish clergy, they were considered to have a poor academic formation, although with an excellent knowledge of their languages, their greatest asset to the parish administration. Taylor (1999, vol. I, p. 124) calculates that only 5% of the priests in the archdiocese of Mexico were Indians in the 1760s. In the last decade of the eighteenth century, the existence of 19 Indians priests and assistants was reported. However, this documentation is incomplete so that it is not possible to perform a

more precise calculation (Taylor 1999, vol. I, p. 141, note 80). We believe that their number should be higher, according to the cases recorded in the archives of the Conciliar Seminar in Mexico.

Confronted with this adverse reality, some Indians priests raised their voices before the highest authorities to demand better opportunities for the indigenous nobility. One of them was the priest Andrés Ignacio de Escalona Arias Acxayacatzib y Temilo, a noble Indian from Tlatelolco, educated at the conciliar seminary of Mexico. In the middle of the 18th century, he solicited the authorities to reopen the Imperial College of Santa Cruz de Tlatelolco, with the purpose of vindicating the poor and ignorant Indians, a situation that made it impossible for them to enjoy " . . . the frankness that the royal piety of His Majesty and His glorious progenitors have granted the natives of these kingdoms, commanding by repeated orders and laws, that they be esteemed as Spaniards in terms of participating in the preeminences granted to the most deserving vassals . . . "(De Escalona 1935, p. 24) Escalona Arias stated that the main purpose of the reopening of the school was to train Indians priests for " . . . the peoples of their countrymen and compatriots and to be able to direct them to the knowledge of our Holy Catholic Faith . . . ". (De Escalona 1935, p. 25)

Escalona also argued that Indians ministers were more suitable because of their command of languages, since the Spanish had a very limited knowledge of them. The clergyman also noted the laws of the late 17th century favoring the Indians and claimed the Lascasian policy: "All the natives of this American kingdom we refer to the defenses, representations and other reasons given by the illustrious Mister Fray Bartolomé de las Casas. (De Escalona 1935, p. 35) This document demonstrates the existence of a literate segment of the indigenous nobility who, as in Peru, sought their vindication in the framework of the Spanish Empire. Along with Escalona, seven other lineage *caciques* from the Valley of Mexico also signed the petition.

A contemporary of Escalona Arias, Julián Cirilo Galicia Castilla, a Tlaxcalan nobleman, promoted on his own the establishment of a new seminary school for noble Indians. Although this school was approved by Charles III, Galicia Castilla was never able to obtain the necessary funding for its opening, so the project finally collapsed (Menegus 2013). Until the end of the colonial era, the indigenous nobility had to accept, with a few exceptions, the secondary place that the Church assigned to it.

There was no lack of Indians clergymen who worked hard to move up in the hierarchy, although with mixed results. A good example is that of the priest José Domingo de la Mota, active in the parishes between the decades of 1730 and 1770 (Luna 2020). A descendant of *caciques* from Mexico City, this person knew how to make the most of the advantages offered by the capital and study in the conciliar seminary and the university, where he obtained his bachelor's degrees in Arts and Theology and even studied law. Together with his good knowledge of Nahuatl, he was soon ordained a priest and started working as an assistant in parishes of the archbishopric. However, Mota was not content with this common destiny; he began to object to the ownership of one of them. It is very probable that his participation in a theology academy, chaired by the eminent theologian and prebendary of Mexico's cathedral, Juan Antonio de Eguiara y Eguren, has earned him recognition that helped him in the opposition for the parishes. He soon obtained those of Tepecoacuilco, Zacualpan, Yautepec, Chalco, Tochimilco, and Tultitlán.

De la Mota undoubtedly excelled, as pointed out by Luna Fierros, for the merits and recognition he achieved, especially for his great interest in ending the social sins of his parishioners, including idolatry. This provoked loud protests from the denounced Indians and even attacks on his physical integrity. However, Mota did not give up and requested the title of canon of Mexico's cathedral, but with no success. Although it was a legitimate aspiration of a priest who was standing out, in practice, rural parish priests were unlikely to achieve that promotion, since the tendency was in favor of clergy doctors who had an academic career in the capital as well as good relations with the high clergy, characteristics Mota lacked. In addition to the personal project of this character, it is

important to emphasize that his path indicated the way for other Indians priests with similar expectations.

Different was the destiny of priest Luciano Paez from Mendoza, whose family support is key to understanding his promotion to a prebend of the high clergy. The career of this bachelor, second son of a cacique from Amecameca, Valley of Mexico, evolved between the end of the Novohispanic period and the early Republican Mexico. Luciano's father was a notable cacique, Luis Paez de Mendoza, a landowner, merchant, and also the governor of Amecameca and the undisputed leader of the local indigenous nobility (Aguirre 2005). Luciano did not inherit the *cacicazgo*; instead, he received full parental support for a church career. He lived in the capital for several years while he studied at the Conciliar Seminary and graduated from university, for which he received support from his older brother, who inherited the title of cacique. In addition, his father owned a house in Mexico, founded a chaplaincy with a capital of 3000 pesos, a rent of 150 pesos, and named Luciano as the chaplain owner (AGN, Fondo Bienes Nacionales, leg. 1374, exp. 22). In 1793, he also inherited important assets in his will:

> It is also my will that my son takes as a way of improvement my silver-plated plywood and my silver-plated stirrup, with more the car adorned with curtains, its six mules and another sawmill that is being tamed and I have just bought, so that it has the decency corresponding to its condition. (AGN, Background Links and Majorities, vol. 262, exp. 1)

Luciano Páez graduated from high school and in the same year he reached the highest order of priesthood (AHAM, Episcopal Fund, book 7, priestly orders 1764–1790). In Mexico, the cacique made friends with Spanish clergymen who could help him in his career, such as Dr. José Nicolás de Larragoiti, priest of Mexico's cathedral and executor of Cacique Páez's will.

Luciano remained linked to his region of birth to care for the economic interests of the cacicazgo since in 1801 he was administering the hacienda of Tlaxomulco (AJP, Archivo Histórico, roll 29). At the same time he continued his ecclesiastical career. In 1813 he was the parish priest of Atlautla (AGN, Fondo Bienes Nacionales, leg. 859, exp. 16) and in 1821 of Ocuituco (AGN, Fondo Bienes Nacionales, leg. 785, exp. 4), both of which were located in the province of Chalco, where his brother, Diego Páez, was the hereditary cacique and who became notable for suppressing insurrections of the Indian people of the region (Herrero 2001, pp. 99–146). Luciano Páez's career reached its peak when he won a canonry from the Insigne y Nacional Colegiata de Nuestra Señora de Guadalupe, which he occupied until his death in 1836, just like the ownership of the chaplaincy that his father had founded 46 years earlier (AGN, Fondo Bienes Nacionales, leg. 1374, exp. 22).

## 6. Some Final Reflections

It is necessary to emphasize the importance of the indigenous nobility for the colonial regime since the 16th century since it became a factor that guaranteed the Crown the political control of the Indians peoples. It is not without reason that from the beginning of New Spain, Charles V recognized the noble Indians as free and loyal vassals, a recognition that was very important in subsequent debates on their inclusion or exclusion in different Hispanic areas. Since then, they were co-participants in the establishment of the colonial regime, and although there was pressure to marginalize them, the Crown provided them with protection and prerogatives that, in the end, could not be omitted by the viceregal authorities. Thanks to this, they received preferential treatment that distinguished them from the rest of the Indians.

The indigenous nobility also maintained a historical memory of their role in the construction of the Spanish regime in the New World, and based on it, demanded their right to government and to the privileges that the regime gave to the Spanish. In this sense, the access of *caciques'* children to major studies, degrees, and the priesthood is part of that permanent aspiration of the nobility to identify with power, honor, and public functions.

This aspiration was objected to by different colonial authorities and social sectors over time but, in spite of this, it did not disappear.

The main power that supported the claims of the noble Indians came from the Crown because, thanks to it, it is possible to explain the forced recognition of Novohispanic actors who were against the educational rise of the Indians. The main impulse came from the government of Charles II at the end of the 17th century. In response to this, the *caciques* with sufficient resources and more familiar with the colonial regime reacted favorably to the monarchic initiative to open spaces in public offices formerly reserved only for Spaniards. Thus, the children of *caciques* began to demand studies in colleges and universities, especially in the conciliar seminaries, where there should be exclusive scholarships for them. The records of the conciliar seminary in Mexico show that, throughout the 18th century, the sons of *caciques* demanded all the scholarships destined to them, or paid to be able to stay and study as interns in that institution. It was mainly during the late colonial period, 1750–1822, that the greatest number of Indians could be found in colleges and universities, an unquestionable sign of the new times lived by the Novohispanic indigenous nobility.

Although the Indians, backed by the Crown, were able to access university and college classrooms, this did not mean that there was social equality. They could meet at the different university spaces with the children of wealthy merchants or powerful auditors or the nephews of high church leaders; they could even share classrooms and religious functions in the chapel, examinations, or poetry contests, but social distances never disappeared.

The most important purpose of the literate Indians was the Church, where they were ordained as priests, mainly looking for the nomination of tenured priests in some parish of their region of origin. This aspiration never ceased to present them with difficulties since, as far as we know, few Indians became priests. The Church in New Spain permitted them only a small promotion: to serve as auxiliaries to the Spanish priests, especially in the administration of the sacraments in indigenous languages, and not in just any priesthood, but rather in those rural towns where almost nobody wanted to go.

The previous pages show, then, how the right to study of the Indians, especially the nobility, was negotiated based on the position of dominance they held in the social structure prior to the arrival of the Spaniards, as well as on their active role in establishing monarchical power over the Indians peoples after the conquest. The former rights of the noble Indians were reflected in their right to study specialized knowledge, in relation to the Spanish strategy of colonization. Thus, the formal rights granted by the Crown were exercised in practice thanks to the mobilization and demands of groups of experienced Indians, supported by some Spanish sectors. However, social distances remained in place due to the resistance of Spanish social groups that kept considering the Indians as conquered and, therefore, lacking the same rights that the former enjoyed.

In the search for social advancement, the Indians were not alone, because since the 17th century other minority groups, such as mulattos and mestizos, also sought access to higher education, especially at the Royal University of Mexico. Although Spanish students continued to be the majority at this university, there are important indications that in the course of the eighteenth century, members of these social groups joined the courses and degrees. The social diversification of the university population was a reflection of the new times in Novo-Hispanic society. A pending task is to investigate the destinies of this type of students and to analyze how studies and degrees helped them to improve their social condition and standard of living.

**Funding:** This research received no external funding.

**Institutional Review Board Statement:** Not applicable.

**Informed Consent Statement:** Not applicable.

**Conflicts of Interest:** The authors declare no conflict of interest.

## Abbreviations

AGN    Archivo General de la Nación, Mexico
AGI    Archivo General de Indias, Seville
AHSCM  Archivo Histórico del Seminario Conciliar de México
AJP    Archivo Judicial de Puebla

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
