# Peer review of "The Indians and Major Studies in New Spain: Monarchical Politics, Debates, and Results"

_socsci, doi:10.3390/socsci10040115_

Round 1

Reviewer 1 Report

The paper engages with a very important and interesting topic by discussing “the issue of the access of the Indians of colonial Spanish America to major studies”, including the university and the clergy. The paper is divided into four sections, which gradually advance and develop the main argument. This is a good discussion, the paper is knowledgeable and well written. The general structure of the paper is methodologically adequate and the reader follows easily the development of the main argument, while the overall analysis is good and convincing.

This is a well-written and readable article, which takes into account both the theoretical and the social/historical dimension of the subject, which proves to be especially useful for the social scientist and the historian. Some minor issues include: a) methodologically, the author uses the concept of the “long term” borrowed from Peter Burke in order to construct her/his argument and draws, additionally, on sources from the political and social history of law. However, post-colonial theory could also be a useful tool and a potential theoretical source for the topic discussed in this paper. Although the bibliography is rich and sufficient for the topic, there should be some references on post-colonial studies, in order to more fully corroborate the author’s argument. To give an example: In p. 3, the author states that “Many European colonizers considered that Indians were little or not at all rational men, who could be enslaved, and others, more condescending, as neophytes, who were not yet Christians and therefore there would be no justification for educating them or allowing them to join the clergy”. A reference to the appropriate sources is needed here; b) in p. 9, just before the end of the third section, the author argues that “The news spread throughout the viceroyalty and the reaction of the Indigenous nobility was immediate, because during the following years more noble Indians began applying for scholarships at other colleges and knocking on the doors of Mexico's Royal University”. A citation is needed here; c) the use of English language is good from a general point of view. However, there are some typographical errors present in the text and some proofreading needs to be done. 

Reviewer 2 Report

The manuscript is an outcome of sound historical research based on archival sources, and presents a well written historiography of an interesting topic that certainly merits the attention given. The language is mostly fluent and idiomatic, excluding the abstract that should be thoroughly edited for language.

Since the paper is part of a special issue, it's fit and linkages to the issue should be improved by writing a short contextualization of what specific themes of the special issue this paper addresses. On the minimum, the paper should address the framing hypothesis presented in the special issue description: "higher education has contributed to the globalization process by connecting some fractions of the social structures of the colonized, and colonizers, through the intersection of higher education access policies and citizenship policies." It would be important that the authors describe how this article contributes to discussions around this theme.

In addition to this, the paper's message would be considerably strengthened by adding some further reflections on these issues to the paper's concluding discussion. On the minimum, the paper should address the theme highlighted in the special issue description: identify groups (including indigenous), that have gone through higher education and how this has impacted on their citizenship status before and after accessing higher education. It would be interesting to link this question with reflections on how education was linked with the colonial rule and how it impacted on the social dynamism in the colonial society.

With these relatively small amendments the manuscript is publishable as part of the special issue "Access to Higher Education in European Colonial Empires: Citizenship, Social Structures and Globalisation".
